# NEWER IS NOT ALWAYS BETTER:
# RETHINKING TRANSFERABILITY METRICS, THEIR PECULIARITIES, STABILITY AND PERFORMANCE

## ABSTRACT

Fine-tuning of large pre-trained image and language models on small customized datasets has become increasingly popular for improved prediction and efficient use of limited resources. Fine-tuning requires identification of best models to transfer-learn from and quantifying transferability prevents expensive re-training on *all* of the candidate models/tasks pairs. In this paper, we show that the statistical problems with covariance estimation drive the poor performance of H-score (Bao et al., 2019) — a common baseline for newer metrics — and propose shrinkage-based estimator. This results in up to $80\%$ absolute gain in H-score correlation performance, making it competitive with the state-of-the-art LogME measure by You et al. (2021). Our shrinkage-based H-score is $3 - 55$ times faster to compute compared to LogME. Additionally, we look into a less common setting of target (as opposed to source) task selection. We demonstrate previously overlooked problems in such settings with different number of labels, class-imbalance ratios etc. for some recent metrics e.g., NCE (Tran et al., 2019), LEEP (Nguyen et al., 2020) that resulted in them being misrepresented as leading measures. We propose a correction and recommend measuring correlation performance against relative accuracy in such settings. We also outline the difficulties of comparing feature-dependent metrics, both supervised (e.g. H-score) and unsupervised measures (e.g., Maximum Mean (Long et al., 2015) and Central Moment Discrepancy (Zellinger et al., 2019)), across source models/layers with widely varying feature embedding dimension. We show that dimensionality reduction methods allow for meaningful comparison across models, cheaper computation ($6\times$) and improved correlation performance of some of these measures. We investigate performance of 14 different supervised and unsupervised metrics and demonstrate that even unsupervised metrics can identify the leading models for domain adaptation. We support our findings with $\sim 65{,}000$ (fine-tuning trials) experiments.

## 1 INTRODUCTION

Transfer learning (TL) is a set of techniques of using abundant somewhat related source data $p(X^{(s)}, Y^{(s)})$ to ensure that a model can generalize well to the target domain, defined as either little amount of labelled data $p(X^{(t)}, Y^{(t)})$ (supervised), and/or a lot of unlabelled data $p(X^{(t)})$ (unsupervised TL). TL is most commonly achieved either via fine-tuning or co-training. Fine-tuning (FT) is a process of adapting a model trained on source data by using target data to do several optimization steps (for example, SGD) that update the model parameters. Co-training on source and target data usually involves reweighting the instances in some way or enforcing domain irrelevance on feature representation layer, such that the model trained on such combined data works well on target data. Fine-tuning is becoming increasing popular because large models like ImageNet (Krizhevsky et al., 2012), Bert (Devlin et al., 2018) etc. are released by companies and are easily modifiable. Training such large models from scratch is often prohibitively expensive for the end user.

In this paper, we are primarily interested in effectively measuring transferability before training of the final model begins. Given a source data/model, a **transferability measure** (TM) quantifies how much knowledge of source domain/model is transferable to the target model. Transferability measures (TMs) are important for various reasons: they allow understanding of relationships between

different learning tasks, selection of highly transferable tasks for joint training on source and target domains, selection of optimal pre-trained source models for the relevant target task, prevention of trial procedures attempting to transfer from each source domain and optimal policy learning in reinforcement learning scenarios (e.g. optimal selection of next task to learn by a robot). If a measure is capable of efficiently and accurately measuring transferability across arbitrary tasks, the problem of task transfer learning is greatly simplified by using the measure to search over candidate transfer sources and targets.

**Contributions** We study both *supervised* and *unsupervised* TMs in the context of FT.
For *supervised* TMs, our contributions are three-fold:

1. We show that H-score, commonly used as a baseline for newer supervised TMs, suffers from instability due to poor estimation of covariance matrices. We propose shrinkage-based estimation of H-score with regularized covariance estimation techniques from statistical literature. We show $80\%$ absolute increase over the original H-score and show superior performance in 9/15 cases against all newer TMs across various FT scenarios.
2. We present a fast implementation of our estimator that is $3 - 55$ times faster than state-of-the-art LogME measure. Unlike LogME, our optimized implementation for our estimator is tractable even for really high-dimensional feature embeddings $\sim 10^5$.
3. We identify problems with 3 other supervised TMs (NCE, LEEP and $\mathcal{N}$LEEP) in target task selection when either the number of target classes or the class imbalance varies across candidate target tasks. We propose measuring correlation against relative target accuracy (instead of vanilla accuracy) in such scenarios.

For *unsupervised* TMs, we outline computational challenges and propose dimensionality reduction methods for better estimation and effective comparison of such measures when the feature dimensions are large and/or different across various source models. We show that with our proposed modifications, even unsupervised TMs can be effective in identifying the best source ImageNet model. Our large set of 65,000 FT experiments with multiple ImageNet models and different regimes generated from CIFAR-100 and CIFAR-10 image datasets shows usefulness of our proposals.

This paper is organized as follows. Section 2 describes general FT regimes and transfer learning tasks. Section 3 discusses supervised TM and addresses shortcomings of the pioneer TM (H-Score) that arise due to limited target data (subsection 3.1). In subsection 3.2 we demonstrate problems with recent NCE, LEEP and $\mathcal{N}$LEEP metrics and propose a way to address them. Section 4 highlights shortcomings of different commonly used unsupervised measures for source selection and proposes alternatives that offer improvements. Finally, Section 5 presents a meta study of all metrics.

## 2    TRANSFERABILITY SETUP

We consider the following FT scenarios based on existing literature.
(i) *Source Model Selection (SMS)*: For a particular target data/task, this regime aims to select the "optimal" source model (or data) to transfer-learn from, from a collection of candidate models/data.
(ii) *Target Task Selection (TTS)*: For a particular (source) model, this regime aims to find the most related target data/task.

In addition, we primarily consider two different FT strategies:
(i) *Linear FT/head only FT*: All layers except for the penultimate layer are frozen. Only the weights of the head classifier are re-trained while fine-tuning.
(ii) *Nonlinear FT*: Any arbitrary layer can be designated as a feature extractor, up to which all the layers are frozen; the subsequent layers include nonlinear transformations and are re-trained along with the head on target data.

## 3    SUPERVISED TRANSFERABILITY MEASURES

**Related Work** Recent literature in transfer learning has proposed computationally efficient TMs. We categorize measures that require target labels as supervised TMs. Inspired by principles in information theory, Negative Conditional Entropy (NCE) Tran et al. (2019) uses pre-trained source model and evaluates conditional entropy between target pseudo labels (source models' assigned labels) and real target labels. Log Expected Empirical Predictor (LEEP) (Nguyen et al., 2020)

modifies NCE by using soft predictions from the source model. Both NCE and LEEP do not directly use feature information, hence they are not applicable for layer selection.

Cui et al. (2018) propose representing each output class by the mean of all images from that class and computing Earth Mover's distance between the centroids of the source classes and target classes. Bao et al. (2019); Li et al. (2021); Huang et al. (2021); You et al. (2021); Deshpande et al. (2021) proposed metrics that capture information from both the (learnt) feature representations and the real target labels. These metrics are more appealing as these can be broadly applicable for models that are pre-trained in either supervised or unsupervised fashion. Li et al. (2021) proposed $\mathcal{N}$LEEP that fits a Gaussian mixture model on the target feature embeddings and computes the LEEP score between the probabalistic assignment of target features to different clusters and the target labels. Huang et al. (2021) proposed TransRate — a computationally-friendly surrogate of mutual information (using coding rate) between the target feature embeddings and the target labels. Bao et al. (2019) introduced H-score that takes into account inter-class feature variance and feature redundancy. You et al. (2021) proposed LogME that considers an optimization problem rooted in Bayesian statistics to maximize the marginal likelihood under a linear classifier head. Deshpande et al. (2021) introduced LFC to measure in-class similarity of target feature embeddings across samples.

Finally, Tan et al. (2021) used Optimal Transport to evaluate domain distance, and combined it, via a linear combination, with NCE. To learn such a measure, a portion of target tasks were set aside, the models were transferred onto these tasks and the results were used to learn the coefficients for the combined Optimal Transport based Conditional Entropy (OTCE) metric. While the resulting metric appears to be superior over other non-composite metrics like H-score and LEEP, it is expensive to compute since it requires finding the appropriate coefficients for the combination. Additionally, our results indicate that both components of the measure seem to be individually sub-optimal in measuring transferability against corresponding supervised and unsupervised measures.

## 3.1 IMPROVED ESTIMATION OF H-SCORE FOR LIMITED TARGET DATA

*H-score* (Bao et al., 2019) is one of the pioneer measures that is often used as a baseline for newer supervised TMs, which often demonstrate the improved performance. It characterizes discriminatory strength of feature embedding for classification:

$$\mathrm{H}(f) = \mathrm{tr}(\boldsymbol{\Sigma}^{(f)^{-1}} \boldsymbol{\Sigma}^{(z)}) \tag{1}$$

where, $d$ is the embedding dimension, $\boldsymbol{f}_i = h(\boldsymbol{x}_i^{(t)}) \in \mathbb{R}^d$ is the target feature embeddings when the feature extractor ($h : \mathbb{R}^p \to \mathbb{R}^d$) from the source model is applied to the target sample $\boldsymbol{x}_i^{(t)} \in \mathbb{R}^p$, $\boldsymbol{F} \in \mathbb{R}^{n_t \times d}$ denotes the corresponding target feature matrix, $Y = Y^{(t)} \in \mathcal{Y} = \{1, \cdots, C\}$ are the target data labels, $\boldsymbol{\Sigma}^{(f)} \in \mathbb{R}^{d \times d}$ denotes the sample feature covariance matrix of $\boldsymbol{f}$, $\boldsymbol{z} = \mathbb{E}[\boldsymbol{f}|Y] \in \mathbb{R}^d$ and $\boldsymbol{Z} \in \mathbb{R}^{n_t \times d}$ denotes the corresponding target conditioned feature matrix, $\boldsymbol{\Sigma}^{(z)} \in \mathbb{R}^{d \times d}$ denotes the sample covariance matrix of $\boldsymbol{z}$. Intuitively, $\mathrm{H}(f)$ captures the notion that higher inter-class variance and small feature redundancy results in better transferability.

We hypothesize that the sub-optimal performance of H-Score (compared to that of more recent metrics) for measuring transferability in many of the evaluation cases, e.g., in (Nguyen et al., 2020), is due to lack of robust estimation of H-Score — see Fig. 1 for a synthetic example showing the non-reliability of empirical H-score over various sample sizes when compared with its population version. Given that many of the deep learning models in the context of TL have high-dimensional feature embedding space — typically larger than the number of target samples — the estimation of the two covariance matrices in H-score becomes challenging: the sample covariance matrix of the feature embedding has a large condition number[1] in small data regimes. In many cases, it cannot even be inverted. Bao et al. (2019) used a pseudo-inverse of the covariance matrix $\boldsymbol{\Sigma}^{(f)}$. However, this method of estimating a precision matrix can be sub-optimal as inversion can amplify estimation error (Ledoit & Wolf, 2004). We propose to use well-conditioned shrinkage estimators motivated by the rich literature in statistics on the estimation of high-dimensional covariance (and precision) matrices (Pourahmadi, 2013). We show that the use of such shrinkage estimators can offer significant gain in the performance of H-score in predicting transferability. In many cases, as our experiments show, the gain is so significant that H-score becomes a leading TM, surpassing the performance of state-of-the-art measures.

---

[1]Condition number of a positive semidefinite matrix $A$, is the ratio of its largest and smallest eigenvalues.

**Proposed Transferability Measure** We propose the following shrinkage based H-score:

$$\mathrm{H}_\alpha(f) = \mathrm{tr}\left(\boldsymbol{\Sigma}_\alpha^{(f)^{-1}} \cdot (1-\alpha)\boldsymbol{\Sigma}^{(z)}\right), \qquad (2)$$

**Estimating $\boldsymbol{\Sigma}_\alpha^{(f)}$** While there are several possibilities to obtain a regularized covariance matrix (Pourahmadi, 2013), we present an approach that considers a linear operation on the eigenvalues of the sample version of the feature embedding covariance matrix. Similar ideas of using well-conditioned plug-in covariance matrices are used in the context of discriminant analysis (Hastie et al., 2001). In particular, we improve the conditioning of the covariance matrix by considering its weighted convex combination with a scalar multiple of the identity matrix:

$$\boldsymbol{\Sigma}_\alpha^{(f)} = (1-\alpha)\boldsymbol{\Sigma}^{(f)} + \alpha\sigma\boldsymbol{I}_d \qquad (3)$$

where $\alpha \in [0,1]$ is the shrinkage parameter and $\sigma$ is the average variance computed as $\mathrm{tr}(\boldsymbol{\Sigma}^{(f)})/d$. The linear op-

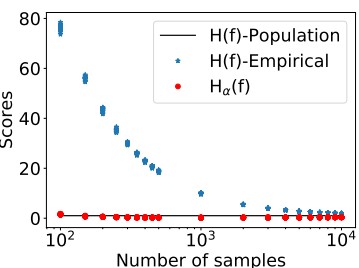

Figure 1: Stability of original H$(f)$ and the shrinkage-based H$_\alpha(f)$ with respect to number of samples. The original H-Score is $\sim 75$ times larger than the population version of the H-Score (estimated with a sample size of $10^6$). In contrast, the shrinkage-based H-Score is significantly more reliable.

eration on the eigenvalues ensures the covariance estimator is positive definite. Note that the inverse of $\boldsymbol{\Sigma}_\alpha^{(f)}$ can be computed for every $\alpha$, by using the eigen-decomposition of $\boldsymbol{\Sigma}^{(f)}$. The shrinkage parameter controls the bias and variance trade-off; the optimal $\alpha$ needs to be selected. This distribution-free estimator is well-suited for our application as the explicit convex linear combination is easy to compute and makes the covariance estimates well-conditioned and more accurate (Ledoit & Wolf, 2004; Chen et al., 2010; Schäfer & Strimmer, 2005).

**Understanding $(1-\alpha)\boldsymbol{\Sigma}^{(z)}$** The scaling factor $(1-\alpha)$ can be understood in terms of regularized covariance matrix estimation under a ridge penalty:

$$1/(1+\lambda) \cdot \boldsymbol{\Sigma}^{(z)} = \mathrm{argmin}_{\hat{\boldsymbol{\Sigma}}} ||\hat{\boldsymbol{\Sigma}} - \boldsymbol{\Sigma}^{(z)}||_2^2 + \lambda||\hat{\boldsymbol{\Sigma}}||_2^2 \qquad (4)$$

where $\lambda \geq 0$ is the ridge penalty. Choosing $\lambda = \alpha/(1-\alpha)$, it becomes clear that $(1-\alpha)\boldsymbol{\Sigma}^{(z)}$ is the regularized covariance matrix.

**Choice of $\alpha$** Ledoit & Wolf (2004) proposed a covariance matrix estimator that minimizes mean squared error loss between the shrinkage based covariance estimator and the true covariance matrix. The optimization with respect to $\alpha$ considers the following objective:

$$\min_{\alpha,v} \ \mathbb{E}[||\boldsymbol{\Sigma}^* - \boldsymbol{\Sigma}||^2] \qquad \text{s.t. } \boldsymbol{\Sigma}^* = (1-\alpha)\boldsymbol{\Sigma}^{(f)} + \alpha v I, \ \ \mathbb{E}[\boldsymbol{\Sigma}^{(f)}] = \boldsymbol{\Sigma}. \qquad (5)$$

where $||\boldsymbol{A}||^2 = \mathrm{tr}(\boldsymbol{A}\boldsymbol{A}^T)/d$. This optimization problem permits a closed-form solution for the optimal shrinkage parameter, which is given by:

$$\alpha^* = \mathbb{E}[||\boldsymbol{\Sigma}^{(f)} - \boldsymbol{\Sigma}||^2]/\mathbb{E}[||\boldsymbol{\Sigma}^{(f)} - (\mathrm{tr}(\boldsymbol{\Sigma})/d) \cdot \boldsymbol{I}_d||^2] \qquad (6)$$

$$\simeq \min\{(1/n_t^2)\sum_{i\in[n_t]}||\boldsymbol{f}_i\boldsymbol{f}_i^T - \boldsymbol{\Sigma}^{(f)}||^2/||\boldsymbol{\Sigma}^{(f)} - (\mathrm{tr}(\boldsymbol{\Sigma}^{(f)})/d) \cdot \boldsymbol{I}_d||^2, 1\}. \qquad (7)$$

where (7) defines a valid estimator (not dependent on true covariance matrix) for practical use. For proof, we refer the readers to Section 2.1 and 3.3 in Ledoit & Wolf (2004). We provide some additional discussion on why same $\alpha$ is used for the two regularized covariance matrices in shrinkage-based H-Score in Supplement Section S2.1.

We provide validation of shrinkage-based estimation of H-Score on synthetic classification data. We generated 1 million 1000-dimensional features with 10 classes using Sklearn multi-class dataset generation function (Pedregosa et al., 2011). Number of informative features is set to 500 with rest filled with random noise. We visualize the original and the population version of the H-score and the shrinkage-based H-Score for different sample sizes in Fig. 1. We observe that the original H-Score becomes highly unreliable as the number of samples decreases. In contrast, the shrunken estimation of H-Score is highly stable and has a small error when compared with the population H-Score.

**Efficient Computation for small target data** For small target data ($C \leq n_t < d$), the naive implementation of H$_\alpha(f)$ can be very slow. We propose an optimized implementation for our shrinkage-based H-Score that exploits diagonal plus low-rank structure of $\boldsymbol{\Sigma}_\alpha^{(f)}$ for efficient matrix inversion

and the low-rank structure of $\mathbf{\Sigma}^{(z)}$ for faster matrix-matrix multiplications. We assume $\boldsymbol{F}$ (and correspondingly $\boldsymbol{Z}$) are centered. The optimized computation of $\mathrm{H}_\alpha(f)$ is given by:

$$\mathrm{H}_\alpha(f) = (1 - \alpha)/(n_t \alpha \sigma) \cdot \left( \|\boldsymbol{R}\|_F^2 - (1 - \alpha) \cdot \mathbf{vec}\left(\boldsymbol{G}\right)^T \mathbf{vec}\left(\boldsymbol{W}^{-1}\boldsymbol{G}\right) \right), \tag{8}$$

where $\boldsymbol{R} = \left[ \sqrt{n_1}\bar{\boldsymbol{f}}_{Y=1}, \cdots, \sqrt{n_C}\bar{\boldsymbol{f}}_{Y=C} \right] \in \mathbb{R}^{d \times C}$, $\boldsymbol{G} = \boldsymbol{F}\boldsymbol{R} \in \mathbb{R}^{n_t \times C}$, $\boldsymbol{W} = n_t \alpha \sigma \boldsymbol{I}_n + \boldsymbol{F}\boldsymbol{F}^T \in \mathbb{R}^{n_t \times n_t}$. The derivation is provided in the Supplement Section S2.3. We make a timing comparison of our optimized implementation of $\mathrm{H}_\alpha(f)$ against the computational times of the state-of-the-art LogME measure and demonstrate $3 - 55$ times faster computation (see Table 6 in Section 5.3).

## 3.2 A closer look at NCE, LEEP and $\mathcal{N}$LEEP measures

Next, we pursue a deeper investigation of some of the newer metrics that are reported to be superior to H-Score and bring to light what appears to be some overlooked issues with these metrics in Target Task Selection (TTS) scenario. TTS has received less attention than Source Model Selection (SMS). To our knowledge, we are the first to bring to light some problematic aspects with NCE, LEEP and $\mathcal{N}$LEEP, which can potentially lead to the misuse of these metrics in measuring transferability. These measures are sensitive to the number of target classes ($C$) and tend to be smaller when $C$ is larger (see Fig. 2[Left]). Therefore, use of these measures for target tasks with *dif-*

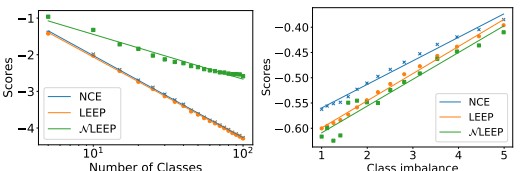

Figure 2: Relation of NCE, LEEP & $\mathcal{N}$LEEP to [Left] number of classes (log-scale) and [Right] class imbalance, $max(n_1, n_2)/min(n_1, n_2)$, for VGG19 on CIFAR100. For [Left], we randomly select 2-100 classes. For [Right], we randomly select 2 classes and vary the class imbalances.

*ferent* $C$ will most likely result in selecting the task with a smaller $C$. However, in practice, it is not always the case that transferring to a task with a smaller $C$ is easier; for example, reframing a multi-class classification into a set of binary tasks can create more difficult to learn boundaries (Friedman et al., 2000). Furthermore, the measures are also problematic if two candidate target tasks have different degree of imbalance in their classes even if $C$ is the same. The measures would predict higher transferability for imbalanced data regimes over balanced settings (see Fig. 2[Right]). However, imbalanced datasets are typically harder to learn. If these measures are correlated against vanilla accuracy, which tends to be higher as the imbalance increases e.g. for binary classification, the measures would falsely suggest they are good indicators of performance. Earlier work did not consider both these aspects and erroneously showed good correlation of these metrics against vanilla accuracy to show dominance of these metrics in TTS with different $C$ (Nguyen et al., 2020; Tan et al., 2021) and imbalance (Tan et al., 2021).

Here, we propose a method to ameliorate the shortcomings of to prevent misuse of these measures, so that they lead to more reliable conclusions. We propose to standardize the metrics by the entropy of the target label priors, leading to the definitions in (9). This standardization considers both the class imbalance as well as number of classes through the entropy of the target label priors.

$$\text{n-NCE} \stackrel{\mathsf{def}}{=} 1 + \text{NCE}/\mathrm{H}(Y), \quad \text{n-LEEP} \stackrel{\mathsf{def}}{=} 1 + \text{LEEP}/\mathrm{H}(Y), \quad \text{n-}\mathcal{N}\text{LEEP} \stackrel{\mathsf{def}}{=} 1 + \mathcal{N}\text{LEEP}/\mathrm{H}(Y). \tag{9}$$

Our proposed normalizations in (9) ensures the normalized NCE is bounded between $[0, 1]$. For proof, see Supplement Section S2.4. n-NCE is in fact equivalent to normalized mutual information and has been extensively used to measure correlation between two different labelings/clustering of samples (Vinh et al., 2010). Given the similar behavior of LEEP and NCE to different $C$ and class imbalance as shown in Fig. 2, we suggest the same normalization as given in (9). However, this normalization does not ensure boundedness of n-LEEP score (and by extension n-$\mathcal{N}$LEEP) in the range $[0, 1]$ as in the case of n-NCE.

For scenarios where candidate target tasks have different $C$, we propose an alternative evaluation criteria (*relative* accuracy) instead of vanilla accuracy — see Section 5 for more details. We provide empirical validation of the proposed normalization to these measures in Table 2 in Section 5.1. We also show that our proposed shrinkage-based H-Score is the leading metric even in these scenarios.

## 4 Unsupervised Transferability Measures

Since most of the existing TMs essentially estimate the difference between source and target distributions (be it just the labels as in LEEP and NCE or between distributions of feature embeddings and labels as in H-score or LFC), one can consider unsupervised discrepancy measures that don't use target labels, but rely only on source and target data feature embeddings. They have been used for regularization in the context of domain adaptation (Li et al., 2020; Zellinger et al., 2019), however they have not been studied for characterizing transferability in the context of FT. It remains to be seen how informative these metrics are as TMs for FT. These metrics are important because they can be used with unlabeled target data when supervised metrics cannot be used.

**Related Work** In early work, Ben-David et al. (2006) proposed $\mathcal{A}$-distance to estimate domain divergence from unlabelled source and target data. This metric can be empirically estimated with an additional model trained to predict the domain of the data (source or target) and uses the domain classification accuracy as a proxy. Training such a model is expensive, requires choosing an architecture and/or tuning hyperparameters and may end up being more expensive than actual TL.

We consider more standard discrepancy measures as unsupervised TMs. Maximum mean discrepancy (MMD) (Long et al., 2015) measures mean differences between distributions in some rich kernel space and has been previously used to detect covariate shift between input distributions (Rabanser et al., 2019). Central moment discrepancy (CMD) (Zellinger et al., 2019) measures differences in mean and higher order moments. Similarly, correlation alignment (CORAL) (Sun & Saenko, 2016) measures the difference in covariance matrices of source and target feature distributions. Kullback-Leibler Divergence (KLD) (Kullback & Leibler, 1951) is a standard divergence metric for two distributions, which can be calculated either by making an assumption on the distributions or using non-parametric estimators. Optimal transport (OT) (Bonneel et al., 2011) considers the optimal energy required to shift distributions from source features to target features. We also consider Wasserstein distance (WD) (Kantorovich, 1939), under normality assumptions for source and target features. Supplement Section S3.2 contains formulas for all the unsupervised TMs.

**Challenges of comparing feature distribution discrepancy across task pairs** Next, we discuss challenges in using unsupervised discrepancy metrics on the feature embeddings of source and target data across model/task pairs in both SMS and TTS scenarios. Given that none of the discrepancy measures (mentioned above) are normalized, direct comparison of these measures across model/task pairs leads to the following challenges:

1. Scale of the features across different source models can be arbitrary, for example ImageNet models with or without Batch Normalization layers.
2. The feature dimension ($d$) across different source models even for the penultimate layer can vary significantly e.g. from 1024 in MobileNet to 4096 in VGG19. Normalization is not straightforward for many metrics such as MMD, KLD etc. Such differences makes source model/layer selection for FT highly problematic.
3. $d$ may be huge. Measuring discrepancy in high-dimensional spaces is both challenging and not well-established due to curse of dimensionality (Rabanser et al., 2019).

**Our proposals to address challenges:** We address the first challenge (outlined above) by standardizing *both* source and target feature embeddings via feature-wise standardization using *source* features' first and second order moments (z-score). See Supplement Section S3.1 for more details. This standardization is in contrast to independent z-score normalization of source and target embeddings, ensuring that measures that consider differences in moments e.g., CMD can effectively capture such information even after standardization.

To address the second and third challenge (outlined above), we propose dimensionality reduction of feature embeddings before computing unsupervised discrepancy measures. We project feature embeddings to a lower $q$-dimensional space, where $q$ is taken to be the same across the competing $K$ models/layers and satisfies: $q \leq \min_{\boldsymbol{f}^{(1)}, \boldsymbol{f}^{(2)}, \dots, \boldsymbol{f}^{(K)}} |\boldsymbol{f}^{(j)}|$ where $|.|$ operator denotes the cardinality of the feature spaces. The dimensionality reduction allows for more meaningful comparison of measures across source/target pairs; this is relevant for source/layer selection. More generally, it also allows for faster and more robust estimate for limited target samples case ($n_t < d$) for linear and nonlinear FT. In the case of nonlinear FT, the intermediate layers of visual and language models have really large $d \sim 10^5$, see Table S2 in Supplement Section S5 for examples.

We consider Principal Component Analysis (PCA) and Gaussian Random Projection (RP). Both use a linear transformation matrix $\boldsymbol{V}$ to derive the transformed features $\hat{\boldsymbol{F}} = \boldsymbol{FV}$; the former derives an optimal orthogonal transformation to capture as much of the variance in the data, while the latter samples components from $\mathcal{N}(0, \frac{1}{q})$ to preserve pairwise distances between any two samples of the dataset. Untrained auto-encoders (AE) are other alternatives that have been used to detect covariate shifts in input distributions by Rabanser et al. (2019). It is not known how sensitive these untrained AE are to the underlying architecture—using trained AE is less appropriate for use in transferability measurement for FT as those maybe more time-consuming than the actual FT. We demonstrate improved correlation performance of unsupervised discrepancy measures with dimensionality reduction in Table 3 in Section 5.1 for TTS and Table 5 in Section 5.2 for SMS.

## 5 Experiments

We evaluate existing TMs and our proposed modifications in various FT regimes and data settings. We draw inspiration from Nguyen et al. (2020) who consider TTS and SMS. The experimental setup highlights important aspects of TMs, e.g., dataset size for computing empirical distributions, covariance matrices and discrepancy measures, number of target classes, and feature dimension etc. Some of these aspects have been overlooked when evaluating TMs, leading to improper conclusions.

Recent work usually considers either supervised or unsupervised domain adaptation. The proposed measures to quantify transferability have mostly relied on the availability of target labels; hence, they have been evaluated in supervised domain adaptation with FT. It has not been previously shown how these measures compare against metrics that only rely on feature distributions e.g. MMD, CMD etc. Therefore, we also provide an empirical evaluation of these measures and demonstrate that even these measures can be effective at identifying the best model to transfer from—this allows for a much broader applicability of the use of these discrepancy measures.

**Fine-tuning with hyperparameter optimization** The optimal choice of hyperparameters for FT is not only target data dependent but also sensitive to the domain similarity between the source and target datasets (Li et al., 2020). We thus tune the hyperparameters for FT: we use Adam optimizer and tune batch size, learning rate, number of epochs and weight decay (L2 regularization on the classifier head). For tuning, we set aside a portion of the training data (20%) and try 100 random draws from hyperparameters' multi-dimensional grid. With this additional tuning complexity, we performed $650 \times 100$ FT experiments. See additional information and motivation in Supplement S4.

**Evaluation criteria** TMs are often evaluated by how well they correlate with the test accuracy after FT the model on target data. Following Tran et al. (2019); Nguyen et al. (2020); Huang et al. (2021), we used Pearson correlation. We include additional results with respect to rank correlations (e.g., Spearman) in Supplement Section S7. We argue that considering correlation with the target test accuracy is flawed in some scenarios. In particular, for TTS, it is wrong to compare target tasks based on accuracy when $C$ is different e.g 5 vs 10 classes. In such a case, task with 10 classes will have a high chance of arriving at lesser test accuracy compared to that for task with 5 classes. In this case, it is more appropriate to consider the gain in accuracy achieved by the model over it's random baseline. Hence we use relative accuracy (for balanced classes): $\frac{\text{Accuracy} - 1/C}{1/C}$. This measure is more effective in capturing the performance gain achieved by the same model in transferring to two domains with different $C$. This also highlights the limitation of NCE, LEEP and $\mathcal{N}$LEEP which are sensitive to $C$ and tend to have smaller values with higher $C$; these measures do not provide useful information about how hard these different tasks when evaluated on the original accuracy scale.

Correlations marked with asterisks (*) in Tables 1, 2, 3, 4 are not statistically significant ($p$-value $> 0.05$). Hyphen (-) indicates the computation ran out of memory or was really slow.

### 5.1 Target Task Selection

We consider Small balanced (S-B) target data, small imbalanced (S-IB) and Large Balance (L-B). See additional details in Supplement Section S6.

**Validation of** $H_\alpha(\boldsymbol{f})$ **against Supervised TMs** We empirically compare the shrinkage-based H-score against the original measure by (Bao et al., 2019) with pseudo-inverse matrix of the sample feature covariance. Table 1 demonstrates $80\%$ absolute gains in correlation performance of $H_\alpha(f)$

over $H(f)$, making it a leading metric in many cases in small target data regimes. Table 2 demonstrates how various supervised TMs perform on TTS when the number of target classes **varies**. $H_\alpha(f)$ dominates the performance in both cases, surpassing all supervised TMs.

Table 1: Correlation comparison of supervised TMs against FT target accuracy. Larger correlations indicate better identifiability as quantified by TM. We compared our proposed $H_\alpha(f)$ against original $H(f)$ and state-of-the-art measures.

| Fine-tuning | Target Data | Model | Regime | $H(f)$ | $H_\alpha(f)$ | NCE | LEEP | $\mathcal{N}$LEEP | TransRate | LFC | LogME |
|---|---|---|---|---|---|---|---|---|---|---|---|
| Linear | CIFAR-100 | VGG19 | S-B | -0.138* | 0.807 | 0.656 | 0.647 | 0.809 | 0.564 | 0.759 | **0.848** |
| | | | S-IB | 0.030* | **0.771** | 0.573 | 0.625 | 0.703 | 0.462 | 0.473 | 0.748 |
| | | ResNet50 | S-B | 0.034* | **0.865** | 0.663 | 0.684 | 0.807 | 0.273 | 0.773 | 0.833 |
| | | | S-IB | -0.103 | 0.785 | 0.560 | 0.569 | 0.699 | 0.437 | 0.518 | **0.819** |
| | CIFAR-10 | VGG19 | S-B | 0.004* | 0.671 | 0.523 | 0.596 | 0.612 | 0.415 | 0.437 | **0.735** |
| | | | S-IB | 0.091* | 0.808 | 0.746 | 0.817 | 0.830 | 0.287 | 0.320 | **0.886** |
| | | ResNet50 | S-B | -0.291 | **0.733** | 0.427 | 0.444 | 0.611 | -0.019* | 0.565 | 0.705 |
| | | | S-IB | 0.170* | **0.893** | 0.656 | 0.708 | 0.752 | 0.279 | 0.005* | 0.832 |
| Nonlinear | CIFAR-100 | VGG19 | S-B | 0.165* | **0.729** | 0.575 | 0.589 | 0.674 | -0.029* | 0.700 | - |
| | | | S-IB | 0.032* | 0.487 | 0.487 | 0.542 | **0.551** | 0.480 | 0.173 | - |

Table 2: Correlation performance of TMs against *relative* accuracy for Large Balanced CIFAR-100 data with different number of classes across target sets.

| Model | $H(f)$ | $H_\alpha(f)$ | NCE | n-NCE | LEEP | n-LEEP | $\mathcal{N}$LEEP | n-$\mathcal{N}$LEEP | TransRate | LogME |
|---|---|---|---|---|---|---|---|---|---|---|
| VGG19 | 0.876 | **0.971** | -0.949 | 0.655 | -0.947 | 0.661 | -0.932 | 0.945 | 0.681 | 0.968 |
| ResNet50 | 0.950 | **0.979** | -0.950 | -0.736 | -0.950 | -0.728 | -0.936 | -0.626 | 0.562 | 0.959 |

**Validating Dimensionality Reduction for Unsupervised TMs** Table 3 shows that unsupervised discrepancy measures greatly benefit from dimensionality reduction (DR). Our findings are summarized below:

- MMD and KLD greatly benefit from dimensionality reduction as they indicate significant improvement in all cases where the correlations hold significance.
- For both CMD and WD, dimensionality reduction improve the correlation in 2/3 cases.
- OT does not gain from DR in estimating transferability.

Table 3: Correlation performance of unsupervised TMs (see Section 4) in FT with/without DR in target task selection for ImageNet Models.

| Target | Model | Regime | DR | CORAL | MMD | CMD | OT | KLD | WD |
|---|---|---|---|---|---|---|---|---|---|
| CIFAR-100 | VGG19 | | None | -0.36 | 0.15 | 0.60 | 0.28 | 0.28 | 0.51 |
| | | S-B | RP | -0.16* | 0.56 | 0.55 | 0.22 | 0.37 | 0.54 |
| | | | PCA | -0.45* | 0.60 | 0.55 | 0.12 | 0.38 | 0.52 |
| | | | None | -0.21* | 0.02* | 0.19 | 0.20 | -0.02* | 0.17 |
| | | S-IB | RP | -0.06* | 0.40 | 0.16 | 0.31 | 0.20 | 0.32 |
| | | | PCA | -0.21* | 0.44 | 0.20 | 0.07 | 0.14 | 0.27 |
| | ResNet50 | | None | -0.36 | 0.02* | 0.54 | 0.38 | -0.21 | 0.50 |
| | | S-B | RP | -0.35 | 0.04* | 0.54 | 0.28 | -0.09 | 0.45 |
| | | | PCA | -0.43 | 0.20* | 0.57 | 0.35 | -0.11 | 0.46 |
| | | | None | -0.08* | -0.13* | 0.24* | 0.25* | -0.40* | 0.15* |
| | | S-IB | RP | -0.04* | -0.20* | 0.17* | 0.24* | -0.16* | 0.12* |
| | | | PCA | -0.07* | -0.15* | 0.24* | 0.21* | -0.24* | 0.18* |

## 5.2 Source Model Selection

We select 9 small to large ImageNet models. We evaluate SMS for FT under small sample setting. We sample 50 images per class from all classes available in the original train split of CIFAR-100/CIFAR-10. We designate 10 samples per class for hyperparameter tuning.

**Validation of $H_\alpha(f)$ against supervised TMs** We highlight that proposed dimensionality reduction techniques in Section 4 are also pertinent for supervised metrics like H-score for SMS with different feature dimensions because the metric is measured in the feature space (despite using target labels). Given that the feature dimensions vary significantly across different models , we apply RP to project to 128-dimensional space ($q = 128$). This allows for more meaningful comparison of H-score across source models and provides the gains of proposed $H_\alpha(f)$ in SMS as well for small samples as given in Table 4, making it again a leading metric in SMS.

Table 4: Correlation of proposed $H_\alpha(f)$ without/with Random Projection (RP) for FT in SMS of ImageNet Models in small data regime constructed from CIFAR-100.

| Regime | Target | $H(f)$ | $H_\alpha(f)$[No RP] | $H_\alpha(f)$[RP] | NCE | LEEP | $\mathcal{N}$LEEP | TransRate | LogME |
|---|---|---|---|---|---|---|---|---|---|
| Linear | CIFAR-100 | -0.190* | 0.024* | **0.859** | 0.825 | 0.839 | 0.852 | -0.204* | 0.705 |
| | CIFAR-10 | 0.276* | 0.277* | **0.939** | 0.938 | 0.936 | 0.938 | 0.311* | 0.923 |
| Nonlinear | CIFAR-100 | -0.108* | 0.125* | 0.879 | 0.967 | 0.976 | **0.977** | - | - |

**Comparison of Unsupervised TMs with Supervised TMs in identifying top model** Table 5 demonstrates which of the metrics can be used to predict the best source model (Top-3 indicates that the best model is in top 3 predictions sorted by this metric). We show that even unsupervised TMs, which can be used both for supervised and unsupervised domain adaptation, exhibit excellent performance in identifying top performing source. Note this performance of unsupervised TMs is with dimensionality reduction (as outlined in Section 4). For linear FT, all of the supervised measures (except for original H-score and TransRate) and unsupervised metrics (except for OT) include the best model in their top 3 predictions. For tuning additional layers, all unsupervised metrics are able to do the same with random projection.

Table 5: Model identification (as measured by Top-3) by supervised/unsupervised measures for FT in SMS of ImageNet Models in small data regime.

| Regime | Target | Supervised TMs | | | | | | | Unsupervised TMs | | | | | | | | | | | |
|--------|--------|------|-------------|-----|------|-------|----------|------|------|----|------|----|------|----|------|----|------|----|------|----|
| | | $H(f)$ | $H_\alpha(f)$ | NCE | LEEP | $\mathcal{N}$LEEP | TransRate | LogME | CORAL | | MMD | | CMD | | OT | | KLD | | WD | |
| | | | | | | | | | None | RP | None | RP | None | RP | None | RP | None | RP | None | RP |
| Linear | CIFAR-100 | ✓ | ✓ | ✓ | ✓ | ✓ | ✓ | ✓ | ✓ | ✓ | ✓ | ✓ | ✓ | ✓ | ✓ | ✓ | ✓ | ✓ | ✓ | ✓ |
| | CIFAR-10 | ✗ | ✓ | ✓ | ✓ | ✓ | ✗ | ✓ | ✗ | ✓ | ✗ | ✓ | ✓ | ✓ | ✗ | ✗ | ✓ | ✓ | ✗ | ✓ |
| Nonlinear | CIFAR-100 | ✗ | ✓ | ✓ | ✓ | ✓ | - | - | - | ✓ | ✗ | ✓ | ✓ | ✓ | ✓ | ✓ | - | ✓ | - | ✓ |

## 5.3 TIMING COMPARISON BETWEEN LOGME AND $H_\alpha(f)$

We empirically investigate the computational times of $H_\alpha(\boldsymbol{f})$ when computed via our optimized implementation in (8). For this exercise, we generate synthetic multi-class classification data using Sklearn (Pedregosa et al., 2011) multi-class dataset generation function that is adapted from Guyon (2003). We investigate different values for number of samples ($n_t$), feature dimension ($d$) and number of classes ($C$). For data generation, we set number of informative features to be 100 with the rest of the features filled with random noise. Table 6 demonstrates a significant computational advantage of $H_\alpha(\boldsymbol{f})$ over LogME. We observe $3 - 55$ times faster computational times. LogME seems intractable both with respect to memory and time for $d \sim 10^5$ as exposed by the nonlinear settings in Table 1 and 4.

Table 6: Timing comparison of LogME and our shrinkage-based H-score. All times are in $ms$.

| $n_t$ | $d$ | $|\mathcal{Y}| = C$ | LogME | $H(f)$ | $H_\alpha(f)$ |
|-------|------|------|--------|-------|------|
| 500 | 500 | 50 | 150 | 95 | **20** |
| 500 | 1000 | 50 | 300 | 200 | **66** |
| 500 | 5000 | 50 | 39100 | 9680 | **1400** |
| 500 | 10000 | 50 | 296000 | 80000 | **5280** |
| 500 | 1000 | 10 | 252 | 170 | **63** |
| 500 | 1000 | 100 | 358 | 202 | **69** |
| 100 | 1000 | 50 | 305 | 248 | **19** |
| 1000 | 1000 | 50 | 333 | 173 | **116** |

## 6 CONCLUSION

We study both supervised and unsupervised TMs in the context of FT. For supervised TMs, our contributions are three-fold. First, we show that H-score measure, commonly used as a baseline for newer supervised TMs, suffers from instability due to poor estimation of covariance matrices. We propose shrinkage-based estimation of H-score with regularized covariance estimation techniques from statistical literature. We show $80\%$ absolute increase over the original H-score and show superior performance in 9/15 cases against all newer TMs across various FT scenarios and data settings. Second, we present a fast implementation of our estimator that provides a $3-55$ times computational advantage over state-of-the-art LogME measure. Unlike LogME, our optimized implementation for our estimator is tractable even for really high-dimensional feature embeddings $\sim 10^5$. Third, we identify problems with 3 other supervised TMs (NCE, LEEP and $\mathcal{N}$LEEP) in TTS (an understudied FT scenario than SMS) when either the number of target classes or the class imbalance varies across candidate target tasks. We propose an alternative evaluation scheme that measures correlation against relative target accuracy (instead of vanilla accuracy) in such scenarios. For unsupervised TMs, we identify challenges with computation and propose dimensionality reduction methods for better estimation and effective comparison of such measures when the feature dimensions are large and/or different across various source models. We show that even unsupervised TMs can be effective in identifying the best source ImageNet model with our proposals. Our large set of 65,000 FT experiments with multiple ImageNet models and different regimes generated from CIFAR-100 and CIFAR-10 image datasets demonstrates usefulness of our proposals. We leave it for future work to explore how predictive various TMs are for co-training regimes (as opposed to FT).

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

## SUPPLEMENTARY MATERIAL

## S1   ACRONYMS

Table S1: List of acronyms used in the paper.

| Terms | Acronyms |
|---|---|
| Transferability Measure | TM |
| Target Task Selection | TTS |
| Source Model Selection | SMS |
| Fine-tuning | FT |
| Dimensionality Reduction | DR |
| Principal Component Analysis | PCA |
| Random Projection | RP |
| Small-Balanced | S-B |
| Small-Imbalanced | S-IB |
| Large-Balanced | L-B |

## S2   SUPERVISED TRANSFERABILITY METRICS

### S2.1   ADDITIONAL DISCUSSION ON SAME SHRINKAGE $\alpha$ FOR THE TWO COVARIANCES IN SHRINKAGE-BASED H-SCORE

The covariance $\mathbf{\Sigma}^{(z)}$ can not be shrunk independently of $\mathbf{\Sigma}^{(f)}$ in the estimation of $\mathrm{H}_\alpha(f)$— the two covariances are coupled by the law of total covariance:

$$\mathbf{\Sigma}^{(f)} = \mathbb{E}[\mathbf{\Sigma}^{(f_Y)}] + \mathbf{\Sigma}^{(z)}. \tag{S1}$$

where $\boldsymbol{f}_Y$ denotes the feature embedding of target samples that belong to class $Y \in \mathcal{Y}$ and $\mathbf{\Sigma}^{(f_Y)} = \mathrm{Cov}(\boldsymbol{f}|Y)$ denotes the class-conditioned covariances. We can write

$$(1-\alpha)\mathbf{\Sigma}^{(f)} = (1-\alpha)\mathbb{E}[\mathbf{\Sigma}^{(f_Y)}] + (1-\alpha)\mathbf{\Sigma}^{(z)},$$

$$\text{i.e, } \mathbf{\Sigma}_\alpha^{(f)} = (1-\alpha)\mathbf{\Sigma}^{(f)} + \alpha\frac{\mathrm{tr}(\mathbf{\Sigma}^{(f)})}{d}\boldsymbol{I}_d = (1-\alpha)\mathbb{E}[\mathbf{\Sigma}^{(f_Y)}] + \alpha\frac{\mathrm{tr}(\mathbf{\Sigma}^{(f)})}{d}\boldsymbol{I}_d + (1-\alpha)\mathbf{\Sigma}^{(z)}. \tag{S2}$$

Comparing (S2) with (S1), we see that the same shrinkage parameter $\alpha$ should be used when using shrinkage estimators, to preserve law of total covariance. The first two terms on the right side in (S2) can be understood as shrinkage of class-conditioned covariances to the average (global) variance. The third term in (S2) (e.g. $(1-\alpha)\mathbf{\Sigma}^{(z)}$) can then be understood as ridge shrinkage as in (4).

### S2.2   CHOICE OF $\alpha$ IN TERMS OF IMPACT ON SHRINKAGE-BASED H-SCORE

Following the synthetic example showing the unreliability of the original H-Score in Section 3.1, we further visualize the effect of using non-optimal values of alpha on the shrinkage-based H-Score. We can see that the shrinkage-based H-Score with optimal shrinkage $\alpha^*$ is much closer to the population version of the original H-Score, especially for smaller sample cases. This validates the use of $\alpha^*$ as computed in equation 7.

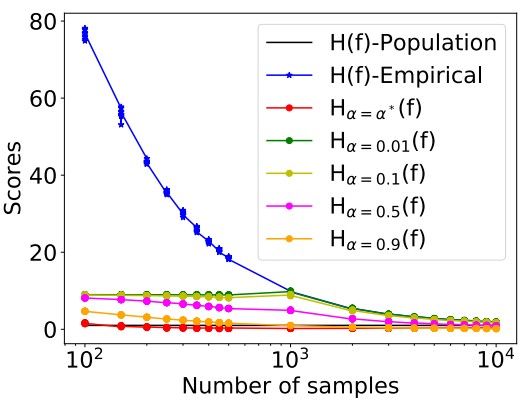

Figure S1: Effect of $\alpha$ on shrinkage-based H-Score.

## S2.3 DERIVATION OF OPTIMIZED IMPLEMENTATION FOR $H_\alpha(f)$

We derive an optimized computation for our proposed shrinkage-based H-score $H_\alpha(f)$ for small target data ($C \leq n_t < d$) as follows:

$$
H_\alpha(f) = \text{tr}\left( \boldsymbol{\Sigma}_\alpha^{(f)^{-1}} \cdot (1-\alpha)\boldsymbol{\Sigma}^{(z)} \right) = \frac{(1-\alpha)}{n_t} \cdot \text{tr}\left( \left( \alpha\sigma\boldsymbol{I}_d + \frac{(1-\alpha)}{n_t}\boldsymbol{F}^T\boldsymbol{F} \right)^{-1} \boldsymbol{Z}^T\boldsymbol{Z} \right),
$$

$$
= \frac{(1-\alpha)}{n_t\alpha\sigma} \cdot \text{tr}\left( \left( \boldsymbol{I}_d + \frac{(1-\alpha)}{n_t\alpha\sigma}\boldsymbol{F}^T\boldsymbol{F} \right)^{-1} \boldsymbol{R}\boldsymbol{R}^T \right),
$$

$$
= \frac{(1-\alpha)}{n_t\alpha\sigma} \cdot \text{tr}\left( \left( \boldsymbol{I}_d - (1-\alpha) \cdot \boldsymbol{F}^T \left( n_t\alpha\sigma\boldsymbol{I}_n + \boldsymbol{F}\boldsymbol{F}^T \right)^{-1} \boldsymbol{F} \right) \boldsymbol{R}\boldsymbol{R}^T \right), \tag{S3}
$$

$$
= \frac{(1-\alpha)}{n_t\alpha\sigma} \cdot \left( \text{tr}\left( \boldsymbol{R}^T\boldsymbol{R} \right) - (1-\alpha) \cdot \text{tr}\left( \boldsymbol{G}^T\boldsymbol{W}^{-1}\boldsymbol{G} \right) \right), \tag{S4}
$$

$$
= \frac{(1-\alpha)}{n_t\alpha\sigma} \cdot \left( \|\boldsymbol{R}\|_F^2 - (1-\alpha) \cdot \textbf{vec}\left( \boldsymbol{G} \right)^T \textbf{vec}\left( \boldsymbol{W}^{-1}\boldsymbol{G} \right) \right), \tag{S5}
$$

where $\boldsymbol{R} = \left[ \sqrt{n_1}\bar{\boldsymbol{f}}_{Y=1}, \cdots, \sqrt{n_C}\bar{\boldsymbol{f}}_{Y=C} \right] \in \mathbb{R}^{d \times C}$, $\boldsymbol{G} = \boldsymbol{F}\boldsymbol{R} \in \mathbb{R}^{n_t \times C}$, $\boldsymbol{W} = n_t\alpha\sigma\boldsymbol{I}_n + \boldsymbol{F}\boldsymbol{F}^T \in \mathbb{R}^{n_t \times n_t}$, (S3) follows by Woodbury matrix identity ($(\boldsymbol{I} + \boldsymbol{U}\boldsymbol{V})^{-1} = \boldsymbol{I} - \boldsymbol{U}(\boldsymbol{I} + \boldsymbol{V}\boldsymbol{U})^{-1}\boldsymbol{V}$) (Max, 1950) and (S4) and (S5) follow by trace properties.

## S2.4 NORMALIZATION OF NCE

NCE (Tran et al., 2019) evaluates conditional entropy between target pseudo labels $Z^{(t)}$ (source model's assigned labels) and actual target labels, as given by:

$$
\text{NCE}(Y^{(t)}|Z^{(t)}) = \frac{1}{n_t} \sum_{i=1}^{n_t} \log p_{Y^{(t)}|Z^{(t)}}(y_i|z_i) \tag{S6}
$$

The conditional entropy of the target labels conditioned on the dummy labels (source model' labels on target data) is:

$$
H(Y|Z) = - \sum_{y \in Y, z \in Z} p_{Y,Z}(y, z) \log p_{Y|Z}(y|z) \tag{S7}
$$

It holds that $0 \leq H(Y|Z) \leq H(Y)$ (see appendix S2.4.1). Negative Conditional Entropy (NCE) is given by $\text{NCE} = -H(Y|Z)$ and it holds that $0 \leq -\text{NCE} \leq H(Y)$. We normalize the NCE as follows:

$$
0 \geq \frac{\text{NCE}}{H(Y)} \geq -1 \tag{S8}
$$

$$
0 \leq 1 + \frac{\text{NCE}}{H(Y)} \leq 1 \tag{S9}
$$

where in (S8) we use the fact that entropy is greater than 0 for any practical classification task.

### S2.4.1 PROOF OF BOUNDS ON CONDITIONAL ENTROPY

$$H(Y|Z) = - \sum_{y \in \mathcal{Y}, z \in \mathcal{Z}} p_{Y,Z}(y,z) \log p_{Y|Z}(y|z) \tag{S10}$$

$$= \sum_{z \in \mathcal{Z}} p_Z(z) \left[ - \sum_{y \in \mathcal{Y}} p_{Y|Z}(y|z) \log p_{Y|Z}(y|z) \right] \tag{S11}$$

$$= \sum_{z \in \mathcal{Z}} p_Z(z) H(Y|Z = z) \tag{S12}$$

$$\geq 0 \tag{S13}$$

where (S13) holds because $H(Y|Z = z) \geq 0$ and holds with equality if and only if $Y$ is a deterministic function of $Z$.

$$H(Y|Z) = - \sum_{y \in \mathcal{Y}, z \in \mathcal{Z}} p_{Y,Z}(y,z) \log p_{Y|Z}(y|z) \tag{S14}$$

$$= - \sum_{y \in \mathcal{Y}, z \in \mathcal{Z}} p_{Y,Z}(y,z) \log \frac{p_{Y,Z}(y,z)}{p_Z(z)} \tag{S15}$$

$$= \sum_{y \in \mathcal{Y}, z \in \mathcal{Z}} p_Y(y) \frac{p_{Y,Z}(y,z)}{p_Y(y)} \log \frac{p_Z(z)}{p_{Y,Z}(y,z)} \tag{S16}$$

$$\leq \sum_{y \in \mathcal{Y}} p_Y(y) \ \log \sum_{z \in \mathcal{Z}} \frac{p_{Y,Z}(y,z)}{p_Y(y)} \frac{p_Z(z)}{p_{Y,Z}(y,z)} \tag{S17}$$

$$= - \sum_{y \in \mathcal{Y}} p_Y(y) \log p_Y(y) \tag{S18}$$

$$= H(Y) \tag{S19}$$

where (S17) holds by Jensen inequality.

## S3 UNSUPERVISED TRANSFERABILITY MEASURES

We denote the target feature embeddings by $\boldsymbol{f}_i = h(\boldsymbol{x}_i^{(t)}) \in \mathbb{R}^d, i \in [n_t]$ and the source feature embeddings by $\boldsymbol{g}_j = h(\boldsymbol{x}_j^{(s)}) \in \mathbb{R}^d, j \in [n_s]$, where $h$ denotes the feature extractor from the source model.

### S3.1 SOURCE AND TARGET FEATURE EMBEDDINGS STANDARDIZATION

We standardize *both* source and target feature embeddings via feature-wise standardization using *source* features' first and second order moments (z-score) as follows:

$$g_{jk} = (g_{jk} - mean(\boldsymbol{g}_{:,k}))/std(\boldsymbol{g}_{:,k}) \tag{S20}$$
$$f_{ik} = (f_{ik} - mean(\boldsymbol{g}_{:,k}))/std(\boldsymbol{g}_{:,k}) \tag{S21}$$

### S3.2 COMPUTATION OF UNSUPERVISED TMS

We list formulae for 6 different distribution discrepancy measures that don't use target labels and instead rely only on source and target feature embeddings.

- *Correlation Alignment (CORAL)* Sun & Saenko (2016) measures discrepancy between the second-order statistics (covariances) of the source and target features as given by:

$$\text{CORAL} = \frac{1}{d^2} \left\| \boldsymbol{\Sigma}^{(f)} - \boldsymbol{\Sigma}^{(g)} \right\|_F^2 \tag{S22}$$

where $\boldsymbol{\Sigma}^{(f)}$ denotes the sample covariance of target features $\boldsymbol{f}$ and $\boldsymbol{\Sigma}^{(g)}$ denotes the sample covariance of source features $\boldsymbol{g}$.

- *Maximum Mean Discrepancy (MMD)* Long et al. (2015) measures the distance of distributions by projecting them into a rich kernel space. In practice, an estimate of the MMD compares the $L_2$ distance between the empirical kernel mean embeddings of the learnt source and target features. Given samples from both source and target feature embedding distributions, we calculate an unbiased estimate:

$$\text{MMD}^2 = \frac{1}{n_s^2} \sum_{i=1}^{n_s} \sum_{j=1}^{n_s} \kappa\left(\boldsymbol{g}_i, \boldsymbol{g}_j\right) - \frac{2}{n_s n_t} \sum_{i=1}^{n_s} \sum_{j=1}^{n_t} \kappa\left(\boldsymbol{g}_i, \boldsymbol{f}_j\right) + \frac{1}{n_t^2} \sum_{i=1}^{n_t} \sum_{j=1}^{n_t} \kappa\left(\boldsymbol{f}_i, \boldsymbol{f}_j\right), \quad \text{(S23)}$$

  where $\kappa$ denotes the radial basis kernel given by $\kappa(a,b) = e^{-\|a-b\|^2/2r}$. To reduce sensitivity of MMD to a particular kernel, we use a combination of radial basis kernels functions with $r \in \{1, 5, 10\}$ as per Long et al. (2015).

- *Central Moment Discrepancy (CMD)* Zellinger et al. (2019) compares means and higher order moments of feature embedding distributions (without projection into kernel space). Given samples from the source and target feature embedding distributions, we only consider the marginal central moments (up to order of 5). The unbiased estimator of the CMD is given by

$$\text{CMD} = \frac{1}{|b-a|} \left\| \mathbb{E}\left(\boldsymbol{g}\right) - \mathbb{E}\left(\boldsymbol{f}\right) \right\|_2 + \sum_{k=2}^{5} \frac{1}{|b-a|^k} \left\| c_k\left(\boldsymbol{g}\right) - c_k\left(\boldsymbol{f}\right) \right\|_2, \quad \text{(S24)}$$

  where $c_k(\boldsymbol{f}) = \mathbb{E}\left[\left(\boldsymbol{f} - \mathbb{E}\left[\boldsymbol{f}\right]\right)^k\right]$, $c_k(\boldsymbol{g}) = \mathbb{E}\left[\left(\boldsymbol{g} - \mathbb{E}\left[\boldsymbol{g}\right]\right)^k\right]$ denote the $k$-th order moment of $\boldsymbol{f}$ and $\boldsymbol{g}$ respectively. $a$, $b$ denotes the joint distribution support of the source and target feature embeddings.

- *Optimal Transport (OT):* This measure considers the optimal energy required to shift distributions from source features to target features. The measure has been well-studied to compare distributions, see Rubner et al. (1998) amongst others. In practice, we solve an optimal transport problem given by

$$\text{OT} = \min_{\boldsymbol{\gamma} \in \mathbb{R}_+^{n_s \times n_t}} \sum_{j \in n_s, k \in n_t} \gamma_{j,k} c_{j,k} \quad \text{s.t.} \quad \boldsymbol{\gamma}\mathbf{1} = \frac{1}{n_s}\mathbf{1}, \quad \boldsymbol{\gamma}^T \mathbf{1} = \frac{1}{n_t}\mathbf{1}, \quad \text{(S25)}$$

  where $c_{j,k} = \left\|\boldsymbol{g}_j - \boldsymbol{f}_k\right\|_2$ denotes the cost of transporting sample feature from source distribution to target distribution. There are entropy regularized variants that include entropy of coupling matrix and efficiently solve the optimization problem with Sinkhorn algorithm (Cuturi, 2013).

- *Kullbach Leibler Divergence (KLD) and Wasserstein Distance (WD) under Multivariate Normal Assumption:* We consider two discrepancy measures under multivariate normal distribution assumptions for source and target features $\mathcal{N}(\bar{\boldsymbol{g}}, \boldsymbol{\Sigma}^{(g)})$ and $\mathcal{N}(\bar{\boldsymbol{f}}, \boldsymbol{\Sigma}^{(f)})$. The closed-form solutions for these two measures are given below:

$$\text{KLD} = \frac{1}{2}\left[\log\frac{|\boldsymbol{\Sigma}^{(f)}|}{|\boldsymbol{\Sigma}^{(g)}|} - d + \text{tr}\left(\boldsymbol{\Sigma}^{(f)^{-1}}\boldsymbol{\Sigma}^{(g)}\right) + \left(\bar{\boldsymbol{f}} - \bar{\boldsymbol{g}}\right)^T \boldsymbol{\Sigma}^{(f)^{-1}}\left(\bar{\boldsymbol{f}} - \bar{\boldsymbol{g}}\right)\right],$$

$$\text{WD} = \sqrt{\left\|\bar{\boldsymbol{g}} - \bar{\boldsymbol{f}}\right\|_2^2 + \text{tr}\left(\boldsymbol{\Sigma}^{(g)} + \boldsymbol{\Sigma}^{(f)} - 2\left(\boldsymbol{\Sigma}^{(g)^{1/2}}\boldsymbol{\Sigma}^{(f)}\boldsymbol{\Sigma}^{(g)^{1/2}}\right)\right)}.$$

## S4 TUNING HYPERPARAMETERS

Current practices for FT typically involve a selection of values for hyperparameters when retraining the model on target data. Given that the target datasets are typically small in the transfer learning scenarios, the typical strategy is to adopt the default hyperparameters for training large models while using smaller initial learning rate and fewer epochs for FT. It has been believed that adhering to the original hyperparameters for FT with small learning rate prevents catastrophic forgetting of the originally learned knowledge or features. Many studies have used fixed hyperparameters (e.g. learning rate, momentum and weight decay, number of epochs) for FT. However, the choice of hyperparameters is not necessarily optimal for FT on target tasks. Earlier work has reported that the performance is sensitive to the default hyperparameter selection, in particular learning rate, momentum (for stochastic gradient descent), weight decay and number of epochs (Mahajan et al., 2018; Kornblith et al., 2019; Li et al., 2020). The optimal choice of these parameters is not only target

data dependent but also sensitive to the domain similarity between the source and target datasets (Li et al., 2020). Therefore, in order to ensure the target task accuracy (against which the correlation of transferability metrics is measured) is optimal, we repeat the FT exercise for 100 trials of hyperparameter settings. We employ Adam for FT experiments and optimize over batch size, learning rate, number of epochs and weight decay (L2 regularization on the classifier head). We select the space of these hyperparameters based on existing literature on FT, e.g. the learning rate is varied in the range $[1e-1, 1e-5]$, the number of epochs between $\{25, 50, 75, 100, 125, 150, 175, 200\}$, the batch size between $\{32, 64, 128\}$ and the weight decay in the range $[1e-6, 1e-2]$.

## S5 FEATURE EMBEDDING LAYERS FOR LINEAR AND NONLINEAR FINETUNING

Table S2: Feature extraction layer in ImageNet models for nonlinear FT. The names are from pretrained ImageNet models in Tensorflow Keras https://keras.io/api/applications/.

| | | Linear Fine-Tuning | | Nonlinear Fine-Tuning | |
|---|---|---|---|---|---|
| | | Feature Embedding Layer | Feature dimensions | Feature Embedding Layer | Feature Dimensions |
| Models | VGG19 | penultimate | 4096 | block3_pool | $28 \times 28 \times 256 = 200,704$ |
| | ResNet50 | penultimate | 2048 | conv2_block3_out | $28 \times 28 \times 256 = 200,704$ |
| | ResNet101 | penultimate | 2048 | conv2_block3_out | $28 \times 28 \times 256 = 200,704$ |
| | DenseNet121 | penultimate | 1024 | pool2_pool | $28 \times 28 \times 128 = 100,352$ |
| | DenseNet201 | penultimate | 1920 | pool2_pool | $28 \times 28 \times 128 = 100,352$ |
| | Xception | penultimate | 2048 | add_6 | $14 \times 14 \times 728 = 142,688$ |
| | InceptionV3 | penultimate | 2048 | mixed4 | $12 \times 12 \times 768 = 110,592$ |
| | MobileNet | penultimate | 1024 | conv_pw_6_relu | $14 \times 14 \times 512 = 100,352$ |
| | EfficientNetB0 | penultimate | 1280 | block3a_activation | $28 \times 28 \times 144 = 112,896$ |

## S6 TARGET TASK SELECTION EXPERIMENTS SETUP

This evaluation regime is motivated by task transfer policy learning in robotics/reinforcement learning. Under this regime, transferability measures can be used to greedily optimize a task transfer policy given a collection of tasks. For instance, a robot has to automatically select which new object to pick up. Given that the robot has learned to pick up a few objects before, it would be beneficial for the robot to optimally select the most transferable source/task object pair and improve it's maneuvering ability throughout the process in a highly efficient manner. TMs can also shed light on the relatedness of different tasks in reinforcement learning setups for better understanding.

We currently evaluate target task selection regime on visual classification tasks with both VGG19 (Simonyan & Zisserman, 2015) and ResNet50 (He et al., 2015) models on subsets of CIFAR-100/CIFAR-10 data under three different dataset regimes following Nguyen et al. (2020). In all 3 cases outlined below, $20\%$ of the samples from the randomly generated subsets is designated as validation set for hyperparameter tuning to find the model with with optimal validation accuracy. We use all examples in the original test set for evaluating out-of-sample accuracy performance on the target data. *Both* training and validation samples in the subsets are used for computation of transferability metrics and we report the correlation of these measures against the (relative) test accuracies for the randomly generated subsets.

- *Small-Balanced Target Data:* We make a random selection of 5 classes from CIFAR-100/CIFAR-10 and sample 50 samples per class from the original train split, out of which we designate 10 samples per class for validation. We repeat this exercise 50 times (with a different selection of 5 classes), fine-tune the model for each selection (100 hyperparameter tuning trials per selection to find finetuned model with optimal validation accuracy) and evaluate performance of those optimal models in terms of test accuracy. We then evaluate rank correlations of TMs across the 50 experiments with random selection of 5 sub-classes.
- *Small-Imbalanced Target Data:* We make 50 random selections of 2 classes from CIFAR-100/CIFAR-10, sample between $30-60$ samples from the first class and sample $5\times$ the number of samples from the second class. This makes for a binary imbalanced classification task. We again measure performance of transferability measures against optimal target test accuracy.
- *Large-Balanced Target Data with different number of classes:* We randomly select 2-100 classes from CIFAR-100 and include all samples from the chosen classes (500 samples per class). This

constructs a range of large balanced dataset target task selection cases. We evaluate correlation of TMs with relative target test accuracy across the variable number of target classes.

## S7  SPEARMAN RANK CORRELATION PERFORMANCE OF SUPERVISED TRANSFERABILITY MEASURES

We present rank correlation performance of all supervised TMs across various FT scenarios (target task selection and source model selection), FT strategies (linear and nonlinear) in various data regimes. Table S3 combines setups in Tables 1, 2, and 4 and presents *Spearman* correlation performance of $H_\alpha(f)$ against supervised TMs. Correlations marked with asterisks (*) are not statistically significant ($p$-value $> 0.05$). Hyphen (-) indicates the computation ran out of memory on 128GB RAM and/or was really slow.

Table S3: Spearman correlation comparison of supervised TMs. Larger correlations indicate better identifiability as quantified by TM. We compared our proposed $H_\alpha(f)$ against original $H(f)$ and state-of-the-art measures. For L-B regimes in the table we correlate against *relative* accuracy. For other rows, we use vanilla accuracy.

| FT scenario | FT strategy | Target Data | Model | Regime | $H(f)$ | $H_\alpha(f)$ | n-NCE | n-LEEP | n-$\mathcal{N}$LEEP | TransRate | LogME |
|---|---|---|---|---|---|---|---|---|---|---|---|
| Target Task Selection | Linear | CIFAR-100 | VGG19 | S-B | -0.19* | 0.77 | 0.67 | 0.67 | 0.81 | 0.56 | **0.86** |
| | | | | S-IB | -0.07* | 0.71 | 0.64 | 0.63 | 0.72 | 0.40 | **0.79** |
| | | | | L-B | 0.96 | **0.97** | 0.50 | 0.44 | 0.95 | 0.91 | 0.96 |
| | | | ResNet50 | S-B | 0.13* | 0.80 | 0.63 | 0.65 | 0.78 | 0.19* | **0.82** |
| | | | | S-IB | -0.10* | 0.76 | 0.57 | 0.58 | 0.69 | 0.41 | **0.81** |
| | | | | L-B | 0.98 | **1.00** | -0.89 | -0.86 | -0.74 | 0.90 | 0.99 |
| | | CIFAR-10 | VGG19 | S-B | 0.06* | 0.57 | 0.49 | 0.49 | 0.55 | 0.30 | **0.65** |
| | | | | S-IB | 0.21* | 0.72 | 0.76 | 0.85 | 0.85 | 0.32 | **0.86** |
| | | | ResNet50 | S-B | -0.31 | **0.60** | 0.28 | 0.29 | 0.51 | 0.03* | 0.59 |
| | | | | S-IB | 0.35 | **0.76** | 0.64 | 0.69 | 0.72 | 0.25* | **0.76** |
| | Nonlinear | CIFAR-100 | VGG19 | S-B | -0.00* | **0.76** | 0.61 | 0.62 | 0.71 | - | - |
| | | | | S-IB | 0.03* | 0.59 | 0.62 | 0.62 | **0.68** | - | - |
| Source Model Selection | Linear | CIFAR-100 | - | Small | 0.30* | **0.88** | 0.83 | 0.83 | 0.80 | 0.35* | 0.83 |
| | | CIFAR-10 | - | Small | 0.07* | 0.88 | 0.93 | 0.92 | 0.92 | 0.07* | **0.95** |
| | Nonlinear | CIFAR-100 | - | Small | 0.052* | **0.96** | 0.93 | 0.93 | 0.93 | - | - |

With respect to Spearman correlations in the table above, our shrinkage-based H-score $H_\alpha(f)$ leads in 7/15 cases and LogME leads in 8/15 cases. In terms of Pearson correlations, $H_\alpha(f)$ leads in 9/15 cases (Tables 1, 2, 4) and LogME leads in 4/15 cases. Additionally, LogME seems to be intractable with respect to memory and computational speed for nonlinear settings where feature dimension is large ($d \sim 10^5$). Our efficient implementation for $H_\alpha(f)$ provides a $3 - 55$ times computational advantage over LogME.

## S8  EXPERIMENTAL CODE AND TYPE OF RESOURCES

We use Tensorflow Keras for our implementation. Imagenet checkpoints (Resnet and VGG) come from Keras https://keras.io/api/applications/. For experiments, we use 2 P100 GPUs per model, 15GB RAM per GPU

