# OpenReview forum: "Newer is not always better: Rethinking transferability metrics, their peculiarities, stability and performance"
_ICLR.cc/2022/Conference — ICLR 2022 Submitted_

### Official Review · Reviewer_7FSx · 2021-10-30

**Correctness:** 3
**Technical Novelty And Significance:** 4
**Empirical Novelty And Significance:** 3
**Recommendation:** 8
**Confidence:** 4

**Main Review:**

Strengths:
1. The paper has a clear transferability setup that makes intuitive senses.
2. The paper improves the state of the arts H-score and some newer metrics by studying both supervised and unsupervised TMs. The paper recognizes the issues with some metrics lie with their sensitivity to the number of classes and imbalanced datasets.

Weakness:
1. The paper only considers the image classification and test on this problem while largely ignoring other tasks. Can the authors try other tasks too?

**Summary Of The Paper:**

The paper studies transferability measure like H-score and shows the instability of H-score due to poor estimation of covariance matrices. The authors propose a new shrinkage-based estimation of H-score and provides a fast implementation for even high dimensional feature embeddings.

**Summary Of The Review:**

The paper is a good paper in my opinion, it tries to address an important problem question in transfer learning: a good transferability metric. It addresses some issues of existent metrics and proposes a fast implementation of the proposed metric. The main concern is that does this metric only work for image classification tasks?

---

### Official Review · Reviewer_M6p2 · 2021-10-31

**Correctness:** 3
**Technical Novelty And Significance:** 2
**Empirical Novelty And Significance:** 2
**Recommendation:** 5
**Confidence:** 3

**Main Review:**

Strengths
- Inside each sections 3 and 4, ideas, intuitions and contributions are well explained and are easy to follow
- While not proposing fundamentally novel ideas, this paper highlights problems of existing methods


Weaknesses
- Hypotheses are not always justified or empirically verified. For instance, the hypothesis that the sub-optimal performance of the H-score for measuring transferability in many of the evaluation cases is due to a lack of robustness in estimating the H-score (page 3) is not empirically demonstrated. What could be a toy example to demonstrate that hypothesis?
- As a whole, the paper could use some restructuring and editing. It’s hard to follow the connections between the supervised and the unsupervised case (both in terms of problems and suggested solutions), how the experiments back up the effectiveness of each solution, whether there is a specific call to action or recommendation for future research on task transferability. For instance, I couldn’t understand where the equations of (11) are used in the experiments.

Questions for Authors
- Table 6 compares the speed of LogMe against shrinkage-based H-score. How does shrinkage-based H score compare to the original H-score in terms of speed?
- Could you elaborate on why Table 5 uses a different metric from Table 1, 2, 3 and 4?
- While I get the use cases for source model selection, could you elaborate on the practical scenarios for target task selection?

Typos Grammar, Presentation Improvements
- It would be worth it to restructure sections 3 and 4 to highlight the problems in both setups and then list the proposed solutions. Something that is confusing is that the experiment section doesn’t reflect the dichotomy of supervised vs unsupervised you introduced in sections 3 and 4.
- Why is Section 2 so small? Could it be integrated into another section instead?



**Summary Of The Paper:**

This paper is interested in task transferability measures (both in the supervised and the unsupervised case). Task transferability measures are useful in quantifying how much knowledge of the source domain or model is transferable to the target model (or target domain in the unsupervised setup). Ultimately, one aims at having effective transferability measures to select the best setup for the task of transfer learning.
The 4 main contributions of this paper are:
- Highlighting instabilities in the computation of the H-score which leads to poor estimate, and proposing to correct these instabilities with regularized covariance estimations, demonstrating an 80% improvement over the original H-score
- Fast implementation that is 3-55 times faster than another state of the art baseline
- Identify problems with other existing transferability measures (NCE, LEEP and N LEEP) and propose to measure correlation against relative target accuracy (instead of vanilla accuracy).
- In the unsupervised setup, propose to use dimension reduction to improve the effectiveness of existing and proposed transferability measures.


**Summary Of The Review:**

While I like that the paper makes a case that existing measures of task transferability have limitations that have been overlooked and that the authors propose simple solutions to fix these problems, I am still confused at the core hypotheses that are not always empirically verified or the actual call to action.

I am willing to raise my score if the authors address the main weaknesses.

---

### Official Review · Reviewer_NC1U · 2021-11-02

**Correctness:** 3
**Technical Novelty And Significance:** 3
**Empirical Novelty And Significance:** 3
**Recommendation:** 5
**Confidence:** 3

**Main Review:**

**Strenghs**

+ The approach proposed in the paper is interesting. In particular, the idea of studying deeply an existing measure rather than proposing yet another new one is a good idea. In particular, the paper builds on robust knowledge in statistics and shrinkage operators to estimate high dimensional covariance.
+ The presentation of the proposed measure as well as its efficient computation is well detailed.
+ Both supervised and unsupervised setting is taken into account.
+ Experimental results are promising.

**Weaknesses and main concerns**

+ The paper could be improved on its structure. In particular, section 3.1 is very dense and some choices could be better motivated, such as instance the choice of using a linear operation on the eigenvalues( equation 3), choice of alpha (equation 5) in terms of their impact on the transferability criteria.
+ Section 3.2 is interesting but what about these measures on SCM ? On different datasets than CIFAR? On different transferability tasks ? For instance on text transfer learning tasks?

**Summary Of The Paper:**

This paper focuses on transferability measures both in a supervised and unsupervised context. In particular, the authors propose a shrinkage-based estimation of H-score in order to correct its instability and discuss the limitations of the other approaches on two different scenarios: source model selection or target task selection. Experiments are done on these tasks on CIFAR.





**Summary Of The Review:**

Interesting paper on robust metrics for measuring transferability having some interesting results and good experimental validation.  Some revisions could improve the paper :
 + Improving the motivations of the different choices in terms of their impact and meaning of the transferability measure.
 + Discuss more the different transferability measures in terms of their meaning in relation with the difficulty and hard settings of transferability tasks.

---

### Decision · Program_Chairs · 2022-01-20

**Decision:**

Reject

**Comment:**

This paper considers transferability measures both in the supervised and unsupervised domain. It identifies instabilities in the way that H-score is computed and proposes to correct the issue with a shrinkage-based covariance estimations. The proposed fix results in 80% absolute gain over the original H-score and makes it competitive with state-of-the-art LogME metric. The new shrinkage-based H-score is much faster to compute.

Reviewers agree that the paper makes interesting and important contributions. In particular, the reviewers appreciate that the paper takes a deeper look at existing metrics and propose valuable fixes instead of proposing yet another new metric. The paper demonstrates depth of statistic knowledge and proposes shrinkage operators to estimate high dimensional covariance.

There are a few shortcomings of the paper, however, that suggests that the paper can benefit of another round of improvement. In particular, the paper is very dense with little motivation. Some of the choices in the paper can be motivated better. For instance, the hypothesis of lack of robustness in estimating H-score is not demonstrated empirically. The reviewers also felt that the paper should extend experiments to other domains beyond image.